# Exploring the Role of Healthcare Personnel in Designing Tuberculosis Infection Prevention and Control Measures in Healthcare Settings: A Scoping Review

**DOI:** 10.3390/ijerph21050524

**Published:** 2024-04-24

**Authors:** Sikhethiwe Masuku, Ramadimetja Shirley Mooa, Mmapheko Doriccah Peu

**Affiliations:** 1TB Platform, South African Medical Research Council, Pretoria 0001, South Africa; 2Nursing Department, University of Pretoria, Pretoria 0002, South Africa; ramadimetja.mooa@up.ac.za (R.S.M.); mpheko.peu@gmail.com (M.D.P.)

**Keywords:** infection prevention and control, guidelines, policies, healthcare personnel

## Abstract

**Background:** Healthcare personnel (HCP) in high TB-burdened countries continue to be at high risk of occupational TB due to inadequate implementation of Tuberculosis Infection Prevention and Control (TB-IPC) measures and a lack of understanding of the context and relevance to local settings. Such transmission in the healthcare workplace has prompted the development and dissemination of numerous guidelines for strengthening TB-IPC for use in settings globally. However, a lack of involvement of healthcare personnel in the conceptualisation and development of guidelines and programmes seeking to improve TB-IPC in high-burden countries generally has been observed. **Objectives:** The aim of this review was to explore the inclusion of HCP in decision-making when designing the TB-IPC guidelines, in healthcare settings. **Methods**: A scoping review methodology was selected for this study to gain insight into the relevant research evidence, identifying and mapping key elements in the TB-IPC measures in relation to HCP as implementors. **Results**: Studies in this review refer to factors related to HCP’s knowledge of TB-IPC, perception regarding occupational risks and behaviours, their role against a background of structural resource constraints, and guidelines’ adherence. They report several challenges in TB-IPC implementation and adherence, particularly eliciting recommendations from HCP for improved TB-IPC practices. **Conclusions**: This review highlights a lack of participation in decision-making by the implementers of the policies and guidelines, yet adherence to TB-IPC measures is anticipated. Future research needs to focus more on consultations with users to understand the preferences from both within individual healthcare facilities and the communities. There is an urgent need for research on the participation of the implementers in the decision-making when developing TB-IPC policies and guidelines.

## 1. Introduction

Healthcare personnel in high tuberculosis (TB)-burdened countries continue to be at high risk of occupational TB due to inadequate implementation of Tuberculosis Prevention and Control (TB-IPC) guidelines and a lack of understanding of the context and relevance to local settings [1]. Healthcare workers all over the world are reported to have latent tuberculosis infection (LTBI), an infection defined based on the cellular immune response to mycobacterial antigens with high exposure rates in primary care and outpatient service [2,3]. In LTBI, healthcare personnel become infected with *Mycobacterium tuberculosis (M. tuberculosis)* that can live in the body without progression to disease [4]. The highest prevalence of LTBI among health workers has been reported in countries with a high burden of tuberculosis in the community than countries with low TB-incident countries [5,6]. This is widely assumed to suggest higher rates of prevalence in the community translate to greater exposure of health workers during provision of care. These assumptions pose a threat to the National Tuberculosis Program in high TB-burdened countries that must focus on prevention measures despite limited resources.

The World Health Organization (WHO) updated the guidelines for tuberculosis infection prevention and control in 2019 [7]. These guidelines are not attempting to create a parallel programme that is exclusive to TB-IPC, instead they emphasise the importance of building integrated well-coordinated multisectoral action to incorporate TB infection control across all levels of care as well as in non-healthcare settings with a high risk of mycobacterium tuberculosis transmission [7]. They lay out general recommendations including practice activities that are crucial for the establishment of a well-functioning Infection Prevention and Control (IPC) programme in all countries [7,8]. 

These core components of IPC programmes form a key part of the WHO strategies to prevent current and future threats; strengthen health service resilience; help to prevent conditions such as healthcare-associated infections, including TB; and combat antimicrobial resistance—by reducing the concentration of infectious droplet nuclei in the air and the exposure of susceptible individuals to such aerosols [9,10,11]. 

Healthcare personnel in high TB-burdened countries continue to be at high risk of occupational TB due to inadequate implementation of TB-IPC and a lack of understanding of the context and relevance to local settings [1]. Healthcare workers all over the world are reported to be infected by LTBI with high exposure rates in primary care and outpatient services [2,3,12]. Such transmission in the healthcare workplace has prompted the development and dissemination of numerous guidelines for strengthening TB-IPC for use in settings globally [13]. However, a growing body of literature points to the lack of involvement [3] of HCP in the conceptualisation and development of guidelines and programmes seeking to improve TB-IPC in high-burden countries generally [14,15]. Many studies conducted in healthcare settings identified non-adherence to TB-IPC measures by healthcare personnel, evident by the poor practices observed and reported [16]. Again, the existing literature focuses on health systems, particularly human resources, and infrastructure that either impede or facilitate implementation of TB-IPC measures in healthcare settings [17]. Although these poor practices coined as non-adherence to TB-IPC by healthcare personnel have been widely documented all over, issues such as why some facilities perform better than others and yet measures are the same are not dealt with. 

A world free from TB can be achieved through the prevention of ongoing transmission, particularly in healthcare settings. Healthcare personnel proving care are not spared of infection, illness, and death from the disease, and of note, nosocomial transmission also apply to patients seeking care. TB prevention and control is one of the key components of the WHO End TB strategy of the second pillar [18]. With appropriate support, healthcare personnel are well positioned to perhaps deliver the change necessary to halt transmission; however, this potential has not been explored.

Views of the implementers on the possibility of them deciding how TB-IPC measures may be tailored to accommodate different settings have not been examined [19]. The aim of this review was to explore the inclusion of the implementers, healthcare personnel, in decision-making when designing the TB-IPC guidelines in healthcare settings. Furthermore, nosocomial transmission is an urgent public health problem, and as such, research needs to go beyond merely looking at noncompliance with suggested measures and seek to address appropriate concepts, generating willingness to be included in decision-making. As various research findings have documented inadequate and noncompliance to TB infection prevention measures all over, this review seeks to generate evidence that could perhaps give implementers the opportunity to own the process, take leadership, and in turn close the gap.

## 2. Methodology

For this study, a scoping review methodology was used. Eligible articles were identified using the framework developed by the Joanna Briggs Institute (JBI) adopted from the work by Arksey and O’Malley (2005) [20] that guides the literature review in the relevance of inclusion criteria. This framework is ideal for collecting, evaluating, and presenting the available research evidence to answer broad topics where many different study designs might be applicable [21]. The framework has five stages consisting of the following: (1) identifying the research question, (2) identifying relevant studies, (3) study selection, (4) charting the data, and (5) collating, summarising, and reporting the results [21]. Undertaking a scoping review for conducting a literature report provides rigor such that the study can be replicable, thereby increasing reliability of the findings [22]. 

Step 1: identifying the research question.

The research question of the study is the key starting point [23]. In this study, we seek to understand the principles of TB-IPC policy development in healthcare settings. The objective of this review is to identify, appraise, and synthesise the evidence on the role of healthcare personnel in policy development in healthcare settings. It is imperative to understand if HCP working in the healthcare settings are involved in TB-IPC policy development. 

Step 2: identifying relevant studies

We reviewed the tittles and the abstract for the records meeting the criteria based on the research question. Electronic versions of the potentially eligible records were retrieved. Further screening of the full text was conducted. Articles were read repeatedly, and the exclusion criteria were applied to reach to the 12 records that were included in this review. Articles were assessed and chosen based on their relevance to the research question rather than methodological rigour. A PRISMA flow diagram giving a detailed account of the strategy used is viewable in Figure 1.

**Concept:** For this review, a literature scoping review, an increasingly adopted approach for reviewing evidence from health-related research was adopted for reviewing evidence on the primary research on TB-IPC in healthcare settings. 

**Content and design:** Studies peer-reviewed and written in English were included in this review. There was no restriction on the date they were published, considering that there is paucity of evidence in this area of research involving HCP in decision-making. Exclusion criteria included review studies, editorials, commentaries, study protocols, conference abstracts, and perspective pieces. Sources included electronic databases, reference lists, and hand searching of key journals.

**Search strategy:** A comprehensive search was carried out in three electronic databases: CINAHL, SCOPUS, and PubMed (NLM). This literature search was performed in the period from November to December 2022. For this review that focused on identifying studies that included HCP in the development of TB-IPC measures as the implementers, we developed a search strategy based on the title and abstract keywords and subject headings to describe our key concepts of health workers consulted/or included in the guidelines/or policies. We applied a filter for articles published in English and did not limit by country. The combination of search terms used is illustrated in Table 1. In addition, studies were searched through cross-referencing and snowballing. Duplicate records were removed before the screening process.

Step 3: study selection

Searches for the three databases were imported into EndNote ™20, Clarivate Analytics, US [24]. A group set of the total records from the databases was created. The next step was to create subgroups of records corresponding to the Prisma flow chart, as shown in Figure 1. Eligible titles and topics were screened after removing all the duplicate records, followed by full-text screening, ending with the records included in the review. Inclusion and exclusion decisions were confirmed at all stages by the second author (M.D.P).

(4) and (5): charting the data, collating, summarising, and reporting the results

Studies included in this review provided insight into the conceptualisation and development of TB-IPC guidelines and programmes seeking to improve TB-IPC in high-burden countries’ implementation and voices of implementers. This primary research sought to occur at the operational level and to identify suggested strategies to deliver sustainable services. The scoping review methodology allows for the inclusion of grey literature and other sources relevant; however, for this review, only peer-reviewed evidence was included to gain evidence as a basis of the phenomenon. Information retrieved from the identified articles included study characteristics (author, year, country, title, aim, methodology) (healthcare personnel inclusion in the policy/guideline conceptualisation), a description of the role of HCP, study outcomes, and brief study findings. The goal of conducting a scoping review is to provide an overview of the available literature to answer the research question; as such, all peer-reviewed studies were included regardless of the quality assessment [20]. Information on the role of HCP was summarised [25].

## 3. Results

A total of 401 records were retrieved, comprising 375 from the three databases, CINAHL, SCOPUS, and PubMed shown in Box 1a. The number of records from snowballing, hand-searched, from other sources was 26. Snowballing, or hand searching, in this context, refers to the strategy of using reference lists or citations in identified papers to identify additional papers [26]. The screened records resulted in a total of 281 records after removing duplicates. There were 196 records excluded after reading the title and abstract. Out of the 85 potentially eligible reports that could meet the selection criteria, 28 were excluded due to full text not available for review. Of the 57 reports that were retrieved, 12 were included in the final review after reading the full text of the articles as illustrated in the PRISMA flow diagram (Figure 1). The included studies were published between 2013 and 2022, and the highest number (*n* = 3) were South African studies, Box 1b.

Box 1Included studies based on year and country of publication.

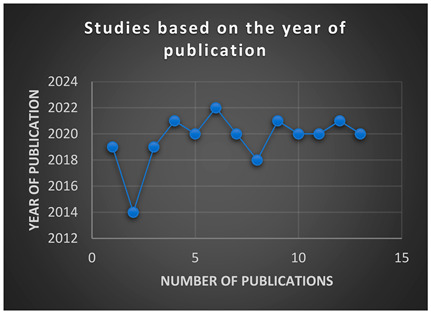



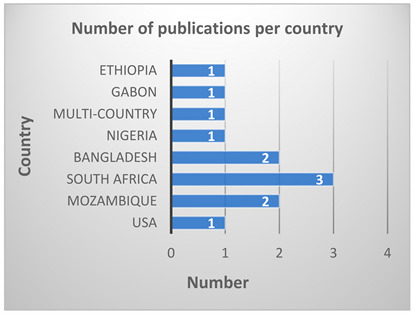

(**a**) 401 studies retrieved from three databases CINAHL, SCOPUS, and PubMed and by snowballing, stratified by year of publication(**b**) Studies included in the analysis were 12, stratified by country of publication

In this review, factors are presented that highlight the role of HCP when looking at the development and implementation of TB-IPC in healthcare settings. However, out of the 12 included studies providing data for the review, only one study specifically focused on the inclusion of the implementers in the decision-making when designing TB-IPC [15]. 

Six qualitative studies focused on the knowledge of TB-IPC among HCP and their perception regarding occupational risks when exposed to occupational hazards, such as TB infection. They explored factors influencing the behaviours toward adoption and compliance with TB-IPC measures in healthcare settings [12,13,15,27]. The other three cross-sectional studies assessed the level of practice and the perceptions of HCP regarding the TB-IPC implementation and adherence at different levels of care in facilities [28,29,30], while the last three described the legal framework and the barriers to policy development and implementation [29,31,32]. A wide range of studies was included in this review, conforming to the scoping review methodology [33]; as there are many options for presenting data in scoping reviews, results are treated as one study in the findings (Table 2). Six qualitative studies in this review refer to factors related to HCP’s knowledge of TB-IPC, perception regarding occupational risks, behaviours, and the role against a background of structural resource constraints, and guideline adherence. They report several challenges in the TB-IPC implementation and adherence, particularly eliciting recommendations from HCWs for improved TBIP practices. In this focus group discussions, challenges emanated from the healthcare system pointing to a lack of clear guidelines and insufficient material/equipment were highlighted [34]. The HCP expressed varied views on challenges in adopting and implementing TB-IPC measures such as a shortage of material, a lack of clear guidelines, insufficient motivation, and inadequate training. While, Zinatsa et al. (2018) [15] identified strategies for improved TB-IPC practices that emphasised comprehensive training, clarity on TB-IPC policy guidelines, and more importantly the active role HCPs can play in infection control as change agents. 

In characterising the perceptions of HCP in practicing protective measures at the work environment, Fix et al. (2019) [35] identified several reasons for non-use and impediments that indicated limitations to the safety culture. Reasons for non-adherence and compliance to TB-IPC provided by the participants included low perception of risks as an impediment to adherence [35]. They indicated that they had no trust in the protocols and safety systems provided and would rather follow personal clinical experiences instead. Meanwhile, Kielmann et al. (2021) [12] highlighted critical gaps relating to sporadic policy changes impeding adoption. In exploring healthcare workers’ knowledge and attitudes and their perceptions towards TB-IPC guidelines [27]. identified gaps in the policies and infrastructure, which impact the implementation. While, Adu et al. (2020) [13] identified a fragmented management of the system, insufficient numbers of healthcare personnel trained in TB-IPC measures, and a lack of recognition of the implementers. The views of implementers were not considered regarding them being involved in decision-making.

The three cross-sectional studies in this review aimed to gain insight and elicit the HCP’s perceptions regarding the TB-IPC implementation and adherence at different levels of care in facilities. Also, the level of practices was determined and assessed against the background of the high level of guideline oversight. The concerns raised included a lack of TB IPC training to nurses and other categories of healthcare personnel tasked with patient care. They were found to be ill-equipped and lacked knowledge and skills required for the adequate implementation of TB-IPC measures. Also, a lack of clear policy directive was determined, and if properly established, this could be of great value in designing an appropriate monitoring and evaluation system for TB-IPC [28]. Healthcare personnel have always been in the frontline of the fight against TB, and the importance of infection control measures has long been undervalued. In this cross-sectional study, Vigenschow et al. (2021) [36] sought to quantify the TB-IPC practices in different healthcare facilities in Gabon. Apparently, there were no national guidelines for TB infection control and PPE was not available for HCP. While Tadesse et al. (2020) [29] reported good knowledge of TB-IPC being associated with optimal implementation of protective measures in hospitals in southern Ethiopia.

The three mixed-methods studies described the legal framework as well as the barriers that hinder the development and implementation of national TB-IPC policies in the health sector. Several barriers impeding the implementation of TB-IPC measures and challenges were reported. Garcia et al. (2020) [31] found that healthcare personnel had limited awareness of their legal rights amongst other barriers, while these elements require attention to protect them from occupational TB. A stronger emphasis on their human rights is needed alongside their perceived responsibilities as caregivers. While Biermann et al. (2020) [37] documented attitudes of TB programme managers related to policy development, implementation, and scale-up, which potentially impact the development and implementation of national policies. We identified underrepresentation of implementors in the development of policies that govern them. In Bangladesh, Nazneen et al. (2021) [32] assessed the status of and barriers impeding the implementation of TB-IPC measures in TB-specialty hospitals and tertiary care hospitals. The study identified lack of knowledge as a major driver for non-adherence to TB-IPC.

**Table 2 ijerph-21-00524-t002:** Characteristics of studies included (n = 12).

Authors	Country	Aim	Methods
Miranda et al. 2014 [34]	Mozambique	To investigate Mozambican HCWs’ perceptions of their occupational TB risk and the measures they report using to reduce this risk and to explore the challenges HCWs encounter while using these TB-IPC measures	Qualitative study design, focus group discussions
Zinatsa et al. [15]	South Africa	This study sought to (1) identify factors influencing TB infection control behaviour at PHC level within a high TB-burden district and (2) in a participatory manner elicit recommendations from HCWs for improved TB infection control	A qualitative case study
Gemmae et al. 2019 [35]	USA	The study sought to characterise perceptions of respiratory protective equipment, identify reasons for use, and examine how work routines might impede or facilitate protocol adherence	Qualitative study design, focus group discussions
Prince et al. 2020 [13]	South Africa	This study sought to elicit perceptions of informed persons within the health system regarding health system barriers to protecting health workers from tuberculosis	Qualitative study
Karina et al. 2021 [12]	South Africa	To examine the role of health workers and managers’ adaptive responses to move the agenda on decentralised DR-TB care forward in pragmatic ways, against a backdrop of structural resource constraints and policy tensions	Qualitative research
Saiful et al. 2022 [27]	Bangladesh	The study examined healthcare workers’ knowledge and attitudes towards TB IPC guidelines and their perceptions regarding the hospitals’ preparedness in Bangladesh	A qualitative exploration
Patrick et al. 2020 [28]	Nigeria	The aim of this study was to determine the levels of TBIC-related knowledge and practices of nurses in Ibadan, South-West Nigeria and their associated socio-demographic factors	Cross-sectional study
Anja et al. 2021 [30]	Gabon	The study was initiated to gain insight into current TBIC practices in different healthcare facilities in Moyen-Ogooué province, in order to properly quantify the dimension of the problem, with the intention to establish baseline data for the future implementation of TBIC measures	Cross-sectional study
Tadesse et al. [29]	Ethiopia	To assess tuberculosis infection control practices and associated factors among healthcare workers in hospitals of Gamo Gofa Zone, southern Ethiopia	Cross-sectional study
Regiane et al. 2020 [31]	Mozambique	This study explores how Mozambique’s legal framework and health system governance facilitate—or hinder—implementation of protective measures in its public (state-provided) healthcare sector	Mixed-methods approach
Biermann et al. 2020 [37]	Multi-country	The aim of this study was to describe attitudes of National TB programme managers related to ACF policy development, implementation, and scale-up in the 30 high-burden countries, which potentially impact the development and implementation of national ACF policies	Mixed-methods study with an embedded design: A cross-sectional survey and qualitative
Nazneen et al. 2020 [32]	Bangladesh	This study aimed to assess the status of and barriers impeding the implementation of TB-IPC measures in TB-specialty hospitals and tertiary care hospitals in Bangladesh	Mixed-methods study

## 4. Discussion

Despite the different foci in these papers, the authors deemed all these subject areas relevant to answering the research question. The reviewed articles presented valuable insight into the relevant research question to be answered. The authors included critical views on factors affecting and influencing the adoption of TB-IPC measures in heath settings. However, there was almost a complete lack of literature exploring the inclusion of the implementers in decision-making, except for one study; Zinatsa et al. [15] reported on the concerns expressed by the healthcare personnel for being left out.

The WHO strongly supports countries demonstrating strong engagement and progress in scaling-up actions to put in place minimum requirements and core components of IPC programmes in every region. However, little progress has been made so far. Only high-income countries are more likely to be progressing their IPC work and are eight-times more likely to have a more advanced IPC implementation status than low-income countries [38].

Middle- to low-income countries that are TB-burdened need to scale up efforts to meet the proposed End TB Strategy to eliminate the disease by 2035. This multiple cause diseases requires multi-sectoral response with political determination to drive down the epidemic at a rapid pace. A synergy of global policies and investments is needed to accelerate TB elimination. Therefore, short-term investments should concentrate on detecting, treating active TB, and most importantly averting new infections. Again, collaborative efforts can be fostered with countries like Brazil, that has been a global reference in TB control with the lowest incident and mortality rates out of the 30 listed TB-burdened countries [39].

The development of the IPC strategic framework is often led by a technical working group in collaboration with various national committees. These were National District Health Systems Committee, National Hospital Coordinating Committee, the Senior Management Committee, and the Ministerial Advisory Committee for Anti-microbial Resistance. Mentioned as key stakeholders in the development of countries’ IPC strategic plans are Government officials, political and healthcare leaders, and policymakers at the ministry of health and other relevant ministries [40], with no recognition to the implementors.

While policy turned to framework gets developed by the appointed technical working teams, a gap remains between policymakers and the implementers who may be in the right position to know what best suits their facility and is fit for purpose [15,41].

Research conducted by Zinatsa et al. (2018) [15] identified several concerns regarding the TB-IPC policies and guidelines at the facility level. Some elements of the guidelines were found to have contradicting statements and to be not suitable for the specific facility environment. Notably, research examined in this review suggests a lack of literature focusing why implementers feel left out in the decision-making on the TB-IPC policy and guidelines. Therefore, research on the participation of the implementers requires more attention from researchers. There is a need for well-designed research studies from lower to middle-income countries, as the available evidence is from developed countries that is difficult to apply broadly.

The existing gaps in TB infection control are linked to barriers in implementing infection prevention measures in many healthcare settings, particularly in high TB-burdened countries. Poor infrastructure, absence of TB infection prevention, and control programmes at the facility level may cause inadequate implementation [42]. It should be mentioned that lack of training and poor managerial involvement in the design and implementation of local TB-IPC policies hinder implementation. Although, COVID somehow brought attention to procurement and the use of personal protective wear such as masks; however, implementation is hampered by a lack of fit testing that is not always performed, thereby offering a sense of false protection to the user [43,44]. Although implementers are aware of policies and guidelines in place, barriers to implementing is negatively influenced by the perception of who needs to take the initiative. Various observers identify a lack of resources and poor infrastructure, but ideas on how implementers can take ownership have not been fully explored. Apart from a lack of focusing on the implementers in decision-making, evidence from the present scoping review offers valuable insight into enablers and barriers to TB-IPC adoption and adherence. Overall, evidence for suboptimal implementation of TB-IPC continues to be overwhelming. However, without involving the implementers, change might be unattainable. Factors to be considered should be the appropriateness of the guidelines to the local context and paying attention to the voices of implementers. Research should therefore move from merely identifying the concerns/reasons of TB-IPC non-adherence to including the voices of the implementers to make the guidelines live, making them a living document continuously moulded in line with the ever-evolving environment to serve the rights and needs of the implementers, i.e., HCP.

## 5. Conclusions

Research on the enablers and barriers to TB-IPC implementation needs to go beyond merely documenting factors affecting and influencing the adoption of TB-IPC measures in heath settings. 

Given the high number of healthcare workers globally that are reported to be infected by LTBI in their line of duty due to the suboptimal implementation of TB-IPC, this warrants being recognised as an area of urgent concern. 

This review highlights a lack of participation in decision-making by the implementers of the policies and guidelines, yet adherence to TB-IPC measures is anticipated. Future research needs to focus more on consultations with users to understand the preferences form both within individual healthcare facilities and the communities. There is an urgent need for research on the participation of the implementers in the decision-making when developing TB-IPC policies and guidelines.

To meet the proposed End TB Strategy efforts to eliminate the disease by 2035, developing countries need to increase resources being channelled towards health systems, strengthening it by allocation of reasonable budgets rather than relying on donor funds. Encouraging and promoting multi-country studies that may bring innovation is key. Optimising the recommended strategies and interventions for TB care and prevention currently not being utilised in developing countries, a first step must be taken.

## Figures and Tables

**Figure 1 ijerph-21-00524-f001:**
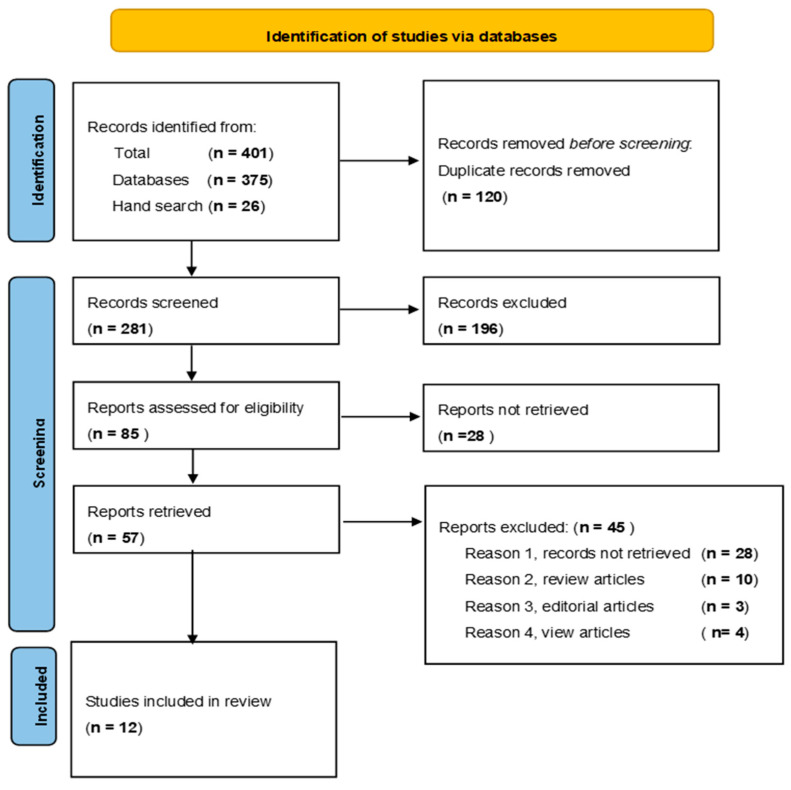
PRISMA flow diagram of study selection and inclusion process.

**Table 1 ijerph-21-00524-t001:** Searches by keywords and document selection.

**Search word/terms:** “(tuberculosis or tb) AND (infection control or infection prevention) AND Policies * AND (healthcare workers *) AND (TB Infection Prevention and Control)”	
**Database**	**Limiters Applied**
MEDLINE	Scholarly (peer-reviewed) journals; linked full text; date of publication: 20100101-20231231; abstract available
CINAHL	Linked full text; abstract available; published date: 20000101-20221231; English language; peer-reviewed; research article; exclude MEDLINE records
SCOPUS	Scholarly (peer-reviewed) journals; linked full text; date of publication: 200020101-20231231; abstract available

* Used to include the widest variety of possible interpretation of the words/concept in the literature.

## Data Availability

The datasets used and/or analysed during the current study are available from the corresponding author on reasonable request.

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
