# Peer review of "Exploring the Role of Healthcare Personnel in Designing Tuberculosis Infection Prevention and Control Measures in Healthcare Settings: A Scoping Review"

_ijerph, 2024, doi:10.3390/ijerph21050524_

Round 1

Reviewer 1 Report

Comments and Suggestions for Authors

The authors nicely framed the manuscript on the role of HCP in designing guidelines and policies for strengthening the TB-IPC in high TB endemic settings. This study contributes to an area of current clinical importance. The authors should address the following comments to improvise the quality of the manuscript.

Minor

1.      Line 15-16: Change this sentence in different words as it is mimicking the text in introduction as well.

2.      Line number 37: Latent TB and subclinical TB are two different stages in TB spectrum. I would recommend removing the term subclinical mycobacterial infection.

3.      Table 2: Change the last column as the text is keep repeating in each referral study.

4.      I would recommend adding the detailed footnotes to the figures.

5.      I felt the discussion is not clearly conveying the practical gaps and the practical issues came up in ground level to implement/improvise the TBIPC.

Comments on the Quality of English Language

Minor editing is required. 

Author Response

Thank you for taking time to review our manuscript. Please see attachment.

Reviewer 2 Report

Comments and Suggestions for Authors

The manuscript "Exploring the role of healthcare personnel in designing Tuberculosis Infection Prevention and Control measures in healthcare settings: A scoping review" is a review and deals with a highly relevant topic that is unfortunately still neglected in the context of global health.

The aim of this scoping review was to explore the inclusion of implementers, healthcare professionals, in decision-making when designing TB-IPC guidelines in healthcare settings.

Recommendations

[1] Introduction

Although the authors highlighted the objective of the scoping review in the Introduction, the text was not clear in this aspect. Therefore, I recommend that the authors further detail the objective of the study.

As a suggestion, therefore not mandatory, the authors should in the Introduction point out some related works in the world that talk about the same theme, this would be an opportune thing, as the authors could better highlight the justification for this study, for example, pointing out gaps in the related works mentioned, which this study would be contributing.

There is an excessive use of acronyms in the introduction.

[2] Methodology

Brazil is a world leader in research and development of actions against tuberculosis. However, I did not find anything substantial in the article that highlighted some work from Brazil. Why this gap?

[4] Discussions

The Scoping Review methodology is very rigorous, but tends to lead authors to a very poor discussion, as they are limited to the findings in the review, when they could develop a more analytical discussion of the scientific findings in this section of the manuscript, therefore not limiting themselves to analysis alone. purely descriptive.

In review articles, the Discussion section should be a space with greater authorship by the authors. Therefore, I strongly recommend that the authors deepen their discussion, and make justified recommendations based on their scientific findings that can contribute to the response to tuberculosis prevention and treatment. What should low- and middle-income countries do to improve the response to tuberculosis?

[5] Conclusions

The conclusion text is very poor, the authors should improve and highlight the importance of eliminating tuberculosis, for example, for the implementation of the 2030 agenda in the context of the Sustainable Development Goals (SDGs).

Author Response

(The authors gave the same response as above.)
